# Identification of New CTX Analogues in Fish from the Madeira and Selvagens Archipelagos by Neuro-2a CBA and LC-HRMS

**DOI:** 10.3390/md20040236

**Published:** 2022-03-29

**Authors:** Àngels Tudó, Maria Rambla-Alegre, Cintia Flores, Núria Sagristà, Paloma Aguayo, Laia Reverté, Mònica Campàs, Neide Gouveia, Carolina Santos, Karl B. Andree, Antonio Marques, Josep Caixach, Jorge Diogène

**Affiliations:** 1Institute of Agrifood Research and Technology (IRTA), Marine and Continental Waters Program, Carretera de Poble Nou, 43540 La Ràpita, Spain; angels.tudo@irta.cat (À.T.); nuria.sagrista@irta.cat (N.S.); palomamaria.aguayo@alumnos.ui1.es (P.A.); laia.reverte@iispv.cat (L.R.); monica.campas@irta.cat (M.C.); karl.andree@irta.es (K.B.A.); jorge.diogene@irta.cat (J.D.); 2Mass Spectrometry Laboratory, Organic Pollutants, IDAEA-CSIC, Jordi Girona 18, 08034 Barcelona, Spain; cintia.flores@idaea.csic.es (C.F.); josep.caixach@idaea.csic.es (J.C.); 3Regional Fisheries Management-Madeira Government, Direção de Serviços de Investigação das Pescas (DSI-DRP), Estrada da Pontinha, 9004-562 Funchal, Portugal; neide.gouveia@madeira.gov.pt; 4Instituto das Florestas e Conservação da Natureza, IP-RAM, Secretaria Regional do Ambiente e Recursos Naturais, Regional Government of Madeira, IFCN IP-RAM, 9050-027 Funchal, Portugal; carolina.santos@madeira.gov.pt; 5Portuguese Institute of Sea and Atmosphere (IPMA), Division of Aquaculture, Seafood Upgrading and Bioprospection (DivAV), Avenida de Brasília, 1449-006 Lisbon, Portugal; amarques@ipma.pt

**Keywords:** ciguatera, fish, liver, ciguatoxin, CBA, LC-HRMS

## Abstract

Ciguatera Poisoning (CP) is caused by consumption of fish or invertebrates contaminated with ciguatoxins (CTXs). Presently CP is a public concern in some temperate regions, such as Macaronesia (North-Eastern Atlantic Ocean). Toxicity analysis was performed to characterize the fish species that can accumulate CTXs and improve understanding of the ciguatera risk in this area. For that, seventeen fish specimens comprising nine species were captured from coastal waters inMadeira and Selvagens Archipelagos. Toxicity was analysed by screening CTX-like toxicity with the neuroblastoma cell-based assay (neuro-2a CBA). Afterwards, the four most toxic samples were analysed with liquid chromatography-high resolution mass spectrometry (LC-HRMS). Thirteen fish specimens presented CTX-like toxicity in their liver, but only four of these in their muscle. The liver of one specimen of *Muraena augusti* presented the highest CTX-like toxicity (0.270 ± 0.121 µg of CTX1B equiv·kg^−1^). Moreover, CTX analogues were detected with LC-HRMS, for *M. augusti* and *Gymnothorax unicolor*. The presence of three CTX analogues was identified: C-CTX1, which had been previously described in the area; dihydro-CTX2, which is reported in the area for the first time; a putative new CTX *m*/*z* 1127.6023 ([M+NH_4_]^+^) named as putative C-CTX-1109, and gambieric acid A.

## 1. Introduction

Ciguatoxins (CTXs) are secondary metabolites produced by marine benthic microalgae (dinoflagellates) members of the genera *Gambierdiscus* [1,2,3,4] and potentially *Fukuyoa* [3,5]. These metabolites accumulate as they are transferred to upper levels in the food webs with concomitant biotransformations and may ultimately reach humans, thereby causing ciguatera poisoning (CP) [6,7]. The CP causes a variety of gastrointestinal, cardiovascular and neurological symptoms [8,9]. Some symptoms can be prolonged during months or even years, severely affecting patients [10,11]. Fatal poisoning cases are rare [12] and it is very difficult to provide a percentage of mortality due to the scarcity of data available from most of CP-prone areas and the nearly universal under-reporting of CP cases. Although, the variability of symptoms and the difficult diagnosis hamper the estimation of CP cases, it is suggested that between 25,000 to 500,000 people are affected each year worldwide [13,14,15].

CTXs are large lipid-soluble polyethers that activate voltage-gated sodium channels (VGSCs) of cells leading to an increase of intracellular sodium and neuronal excitability [16]. CTXs are tasteless, colourless, odourless and they cannot be eliminated by any food preservation or cooking technique [17]. Hence, prevention of CP relies on avoiding consumption of seafood tainted with CTXs and their analogues. More than 400 species of fish [18] and also a few shellfish species have been related to CP cases [19]. Top predators and large fish seem to have the highest CTXs concentrations and are involved more often in CP outbreaks than herbivorous fish [20,21,22]. Although in certain locations in the Pacific region, herbivorous fish are the main concern in CP outbreaks [23].

Historically, CP was confined to circumtropical areas including the Caribbean Sea, tropical islands of the Pacific Ocean and regions in the Indian Ocean (35° N, 35° S) [6]. Cases of CP outside these areas resulted from consumption of imported fish or the return of sick travellers from endemic areas where they had been poisoned [24]. Currently, CP cases in temperate areas that have been related to autochthonous fish have increased [19]. Among the suspected causes, climate change could have influenced the migratory pattern of fish and the distribution of the CTX-producing microalgae [25]. In Europe, an area with recent CP cases caused by consumption of local fish is the region of Macaronesia, which is a collection of different Archipelagos in the North Atlantic Ocean: Cape Verde, Canary Islands, Selvagens, Madeira, and Azores. Since 2004, several species of the genus *Gambierdiscu*s have been reported in this area [26,27,28,29] and their CTX-like toxicity and production of CTX analogues have been described [28,30,31,32]. In fact, *Gambierdiscus excentricus* was found in the Canary Islands [28] and the Madeira Archipelago [26], and it is considered after *G. polynesiensis* as one of the species producing the highest amounts of CTXs [3,28,31,32].

The first CP case in the Macaronesian region was reported in 2004, in the Canary Islands (Spain) [33], and since then, numerous cases have been reported. The Canary Islands government implemented an official control program based on the CTX-like toxicity of the classified ciguateric fish species with specific weights. In the Canary Islands, these fish are analysed for CTX-like toxicity with the neuroblastoma cell-based assay (neuro-2a CBA) [34]. This method has been widely used to detect and quantify CTX-like toxicity [32,35,36,37,38,39,40]. Presently, the detection limit of the neuro-2a CBA is below the safety level proposed by the Food and Drug Administration (FDA) of the USA [41], which is also being contemplated by the European Food Safety Authority [42]. Contrarily, in the Madeira and Selvagens Archipelagos (Portugal), an official control to test the CTX-like toxicity of fish has not been implemented. However, two precautionary measures have been established including a fishing ban around the Selvagens Islands to a depth of 200 m and ban on catching amberjacks of more than 10 kg. To date, in the Madeira and Selvagens Archipelagos, mainly fish from the reserve of the Selvagens Islands has been involved in CP. However, also fish from Madeira were involved in CP outbreaks of 2015 and 2016 [43]. Autochthonous fish that have been related to CP cases in the Madeira and Selvagens Archipelagos are *Pagrus pagrus*, *Seriola* spp. *Ephinephelus marginatus* and *Bodianus scrofa* [43]. Additionally, in the Selvagens Islands (Selvagem Grande and Selvagem Pequena) and Madeira, CTX analogues have been confirmed in fish. The isoforms CTX1B, CTX3C and a CTX analogue have been detected by ultraperformance liquid chromatography-mass spectrometry (UPLC-MS) [44] and C-CTX1 was detected by liquid chromatography with tandem mass spectrometry detection (LC-MS/MS) [45,46,47,48] and by high-resolution mass spectrometry (LC-HRMS) [49]. The Madeira and Selvagens Archipelagos are not the most northern region in the Macaronesia area where CTXs have been found in marine organisms, since two starfish species (Fam. Asterozoa) from the Azores Archipelago, which is located between 37° and 40° N latitude and 24° and 32° W longitude, presented CTX analogues by Ultra Performance Liquid Chromatography-Mass Spectrometry-Ion Trap-Time of Flight (UPLC-MS-IT-TOF) and UPLC-MS [50].

Detection of CTXs is difficult basically because of the diversity of molecular congeners and a lack of certified CTXs standards and reference materials. At the moment, there are up to 30 known analogues of CTXs. According to FAO and WHO [25], thanks to advances in structural elucidation, there is a proposal to classify CTXs into four groups: the ciguatoxin 4A group (including CTX4A and derivatives), the ciguatoxin 3C group (including CTX3C and derivatives), the Caribbean ciguatoxin group (including C-CTX1 and derivatives), and the Indian ciguatoxin group (including I-CTX1 and derivatives). When defining the CTX analogues present, the extraction procedure and the detection methodologies have to be carefully considered as there is no consensus method for detection of all CTXs, probably due to the wide range of fish species, matrices and CTX analogues [51]. In the present study, seventeen fish specimens from nine species were obtained from the Madeira and Selvagens Archipelagos and they were extracted with two different extraction methods. Further, the evaluation of potential CTX compounds was performed using two different approaches. First, a screening method using the neuro-2a CBA for CTX-like toxicity was applied. In a second step, the identification of CTX compounds in the most toxic samples was performed using LC-HRMS analysis. 

## 2. Results

### 2.1. Identification of Fish by COI Sequencing 

The approach of identification of fish species using COI sequencing followed the protocol indicated in Kochsiuz et al. [52]. The premise for this approach is that the existing databases should provide reference material for comparison of unknowns. The results of the samples analysed in this study were not in conflict with the morphological identification. However, for the two species of moray eel*, Muraena augusti* and *Gymnothorax unicolor*, there was no reference sequence for COI in GenBank. Phylogenetic analyses (Appendix A) place these species into their respective genera, but with long branches separating them from other species. The COI sequence data of this work is very likely the first obtained for these two species.

### 2.2. CTX-like Toxicity Evaluation by Neuro-2a CBA

As it was expected, after the exposure of neuro-2a cells to several concentrations CTX1B in O/V− conditions cell viability was not affected. Otherwise, the exposure of neuro-2a cells to CTX1B in O/V+ conditions affected to cell viability. Figure 1a shows a dose-response of standard curve using CTX1B. Intra-assay variability of the relative standard deviation of controls in conditions O/V− and O/V+ controls ranged 1.7 to 5.8% and 2.4 to 6.2%. None of the fish muscle samples extracted using protocol-A exhibited CTX-like toxicity with the neuro-2a CBA. The inter-assay mean of LOQ (±standard deviation, SD) of fish muscle for protocol-A was 0.019 ± 0.014 µg CTX1B equiv·kg^−1^. After the negative results for CTX-like toxicity, the muscle and liver of the fish specimens were extracted using protocol-B and their CTX-like toxicity was evaluated with the neuro-2a CBA. Table 1 shows the CTX-like toxicity of fish obtained with the neuro-2a CBA using extracts produced with protocol-B. Thirteen liver samples and four muscle samples presented CTX-like activity above the LOQ. The inter-assay mean of LOQ (±SD) for protocol-B was 0.025 ± 0.038 µg CTX1B equiv·kg^−1^ for fish muscle and 0.040 ± 0.044 µg CTX1B equiv·kg^−1^ for fish liver. The CTX-like toxicity in muscle samples ranged from 0.034 to 0.083 µg CTX1B equiv·kg^−1^ and for liver samples ranged from 0.010 to 0.270 µg CTX1B equiv·kg^−1^. A dose-response curve of liver of the specimen with code 35 (the most toxic extract) is shown in Figure 1b.

The specimens that presented CTX-like toxicity in muscle were four moray eels of 3 species, *Muraena helena, M. augusti* and *G. unicolor*. The most toxic muscle sample was from a *M. helena* specimen (code 39), with 0.083 ± 0.014 µg CTX1B equiv·kg^−1^. For those fish presenting toxicity in muscle, their corresponding liver exhibited CTX-like toxicity as well. Nonetheless, some fish with toxicity in liver did not present toxicity in muscle tissue (see Table 1). The CTX-like toxicity was higher in the liver than in the muscle for all fishes. The other fishes with signals of CTX-like toxicity in liver but not in muscle were: five moray eels (*M. helena, M. augusti* and *G. unicolor*), one comb grouper (*Mycteroperca fusca*)*,* one barred hogfish (*B. scrofa*), one Bermuda sea chub (*Kyphosus sectatrix*) and one triggerfish (*Balistes capriscus*). The highest CTX-like toxicity was estimated for one liver of a moray eel of the species *M. augusti* (code 35), with 0.270 ± 0.121 µg CTX1B equiv·kg^−1^. The relation between the CTX-like toxicity quantifications of *G. unicolor* and the weight of fish is shown in Appendix A. Considering all four positive individuals, the linear relationship was assessed between CTX-like toxicity and weight, a potential correlation was observed increasing the CTX-like concentration when the weight increased and a positive linear correlation between these two variables was observed with an R^2^ = 0.79.

### 2.3. Identification of CTXs by LC-HRMS Analysis

After the neuro-2a CBA screening analyses, the extracts of the four most toxic livers corresponding to three *G. unicolor* (codes 25, 26, 27) and one *M. augusti* (code 35) specimens and the corresponding fish muscle samples of fish 26 and 27 were analysed by LC-HRMS. The spectra of CTXs were dominated by [M+H]^+^, [M+NH_4_]^+^ and [M+Na]^+^, which were used for confirmatory purposes. According to EU Commission SANCO/12571/2013 guidance document, that included requirements for identification of analytes and confirmation of results by LC-HRMS for pesticide residues analysis in food and feed, these three signals were used as diagnostic ions. Each signal by LC-HRMS supposes two identification points. As there is no similar document for the analysis of toxins, this document has been chosen as a reference. The same criteria are widely used in LC-HRMS analyses of different families of compounds.

All CTX analogues were identified using their measured *m*/*z*, mass accuracy (ppm), ring double bond equivalents (RDBEs) and the mono-isotopic pattern (M+1 ion) of the main adduct signal; all results are summarized in Appendix A. Structure elucidation of unknown molecules by mass spectrometry is a challenge despite advances in instrumentation. The first crucial step is to have an accurate mass measurement with adequate resolution and accuracy to obtain correct elemental compositions. In order to constrain the thousands of possible candidate structures, rules such as RDBE, nitrogen and seven heuristic rules (including isotopic pattern), need to be developed to select the most likely and chemically correct molecular formulas. All CTX have a characteristic isotopic pattern (fingerprint), with a high abundance M+1 signal, mainly due to the high number of carbons in all CTX molecules. M+1/M ion ratio, is a robust criterion that for all described CTXs is a theoretical value of 0.6–0.7. Additionally, the RDBE term measures the degree of unsaturation of the molecule.

Ciguatoxins are a class of polycyclic polyethers with a characteristic value for RDBE. The type of adduct, [M+H]^+^, [M+NH_4_]^+^ or [M+Na]^+^, also influences the value of this RDBE. Exact mass, isotopic pattern and RDBE theoretical and experimental should be coincident.

The analogue C-CTX1 (C_62_H_92_O_19_) was identified in all four liver samples at a retention time of 7.59 ± 0.05 min by LC-HRMS (Figure 2a,b). The identification of this CTX analogue was identified detecting *m*/*z* 1141.6305 [C_62_H_92_O_19_+H]^+^. The mass accuracy of [M+H]^+^ matched ppm ≤ 3.9, RDBEs was in all cases 16.5 and the monoisotopic pattern (M+1 ion) ratio of the main signal was between 0.55–0.69 (See Appendix A).

A putative CTX analogue corresponding to dihydro-CTX2 (C_60_H_84_O_18_) or structural isomer was identified in four samples (muscle and liver of fish 26, muscle of 27 and liver of 35) at a retention time of 6.62 ± 0.08 min by LC-HRMS (Figure 2c,d). To our knowledge, this signal has not been described in any previous study. The identification of this CTX analogue was identified detecting *m*/*z* 1115.5611 [C_60_H_84_O_18_+Na]^+^. The mass accuracy of [M+Na]^+^ matched ppm ≤ 5.7, RDBEs was in all cases 18.5 and the monoisotopic pattern (M+1 ion) ratio of the main signal was between 0.58–0.65 (See Appendix A).

Gambieric acid A (C_59_H_92_O_16_) or structural isomer was identified in two samples (liver of fish 35 and muscle of fish 27) at a retention time of 3.93 ± 0.10 min by LC-HRMS (Figure 2e,f). The identification of gambieric acid A was identified detecting *m*/*z* 1057.6458 [C_59_H_92_O_16_+H]^+^, 1074.6726 [C_59_H_92_O_16_+NH_4_]^+^ and 1079.6278 [C_59_H_92_O_16_+Na]^+^. The mass accuracy of [M+H]^+^ matched ppm ≤ 3.5, RDBEs was 13.5 and the monoisotopic pattern (M+1 ion) ratio of the main signal was 0.59 (See Appendix A). Additionally, the mass accuracy of [M+NH4]^+^ and [M+Na]^+^ matched ≤2.4 ppm and ≤4.7 ppm, respectively. 

Finally, a putative new CTX analogue, with a later retention time than CTX1B (reference retention time: 6.39 min) on the C_18_ column and with a *m*/*z* 1127.6023 [M+NH_4_]^+^ and 1132.5577 [M+Na]^+^ was tentatively detected in all samples (four liver and two muscle samples) at 7.06 ± 0.05 min by LC-HRMS (Figure 2g,h). The isotopic patterns of both signals [M+NH_4_]^+^ and [M+Na]^+^ are identical to the characteristic isotopic pattern of CTXs (M+1/M = 0.55–0.7). The experimental M+1/M ratio was between 0.53–0.66 (see Appendix A). Additionally, the mass accuracy of [M+NH_4_]^+^ and [M+Na]^+^ matched ≤8.5 ppm and ≤6.0 ppm, respectively. 

The observed signals, however, do not agree with any CTX of those already described in the literature. According to molecular formula proposed by the Xcalibur software, unsuitable RDBE data were found, and the nitrogen rule did not fit. Complementary techniques such as MS^n^ and RMN should be applied in order to obtain the structural characterization of this substance.

## 3. Discussion

### 3.1. Identification of Fish by COI Sequencing 

Genetic analysis of fish species is a useful tool for establishing records and subsequently implement risk assessment for fish consumption. Many factors can confound a morphological identification such as the state of conservation of the sample, whether the sample consists of a whole fish or merely a fillet, the existence of sexual dimorphism between genders, the ontogenetic differences between juveniles versus adults, and environmental factors affecting size and/or coloration. Molecular analysis can identify species even from partially digested stomach contents. However, the utility of this method relies on a well-annotated database for comparison that includes region-specific haplotypes. In the case of teleost fish species, many intensely studied species that are of commercial interest have full genomes in international databases like EMBL and GenBank. For many wild species, collection of COI sequences is still incomplete. Such seems to be the case for the genera *Gymnothorax* and *Muraena*. This may be true generally for many benthic and deep-water fish species and more broadly for all non-commercially exploited species that are not so easily brought to land for study. Most of the species identified morphologically were confirmed using phylogenetic analyses except for *G. unicolor* and *M. augusti*. Although the long monophyletic branches for each of these sequences obtained suggest that morphological identification is correct, these are the first sequences for COI obtained for these species.

### 3.2. Toxin Analysis by CBA and LC-HRMS Analysis

After all the negative results of CTX-like toxicity with the neuro-2a CBA for all extracted muscle samples using protocol-A demonstrated that this protocol may be misleading. Muscle and liver of the seventeen fish specimens were extracted for toxin analysis using protocol-B and the CTX-like toxicity of these fish was re-evaluated with the neuro-2a CBA. In contrast to the results using protocol-A, four muscle samples extracted using protocol-B showed CTX-like toxicity. This result suggests that protocol-B is more appropriate to detect CTXs in fish muscle. Previously, several studies had used the extraction protocol-A [36,53] and the results were CTX-like positive, however protocol B was not evaluated in these previous studies. The main difference between both protocols is the order of the liquid-liquid extraction procedures. While in protocol A the hexane cleaning is performed prior to the diethyl ether extraction, in protocol B the diethyl ether extraction is performed prior to the hexane cleaning. Nevertheless, according to the results of the present study, detection of CTX-like toxicity can be enhanced using protocol-B. 

For the fish tested with codes 14, 38, 28, 6, 1, 13 and 37, the toxicity value for the livers was below 0.06 µg CTX1B equiv·kg^−1^ and CTX-like toxicity was not detected in muscle. There were two out six fish that presented CTX-toxicity in liver (>0.06 µg CTX1B equiv·kg^−1^), but CTXs were not detected in muscle. Thus, we can conclude that it is important to survey different tissues (particularly liver tissues) to establish whether or not a given fish contains measurable CTXs concentrations. 

In this study, the maximum difference on CTX-like toxicity between liver and muscle was five times higher in liver than in muscle. Thus, this maximum difference for CTX-like toxicity could be even higher. The higher toxicity found in liver as compared to muscle is in accordance with Chan et al. [54], where the highest CTX concentration in the liver was nine times higher than in muscle. However, most articles evaluate CTXs only in muscle [46,55,56,57,58]. This approach may be justified by the fact that muscle is the most consumed tissue. However, in some species and regions, the viscera are also consumed. The evaluation of liver, in addition to muscle, may improve risk assessment studies and may be important as a warning approach in areas where CP may be increasing in intensity and is not yet affecting public health. On the other hand, when addressing evaluation of toxins in the environment and in different species of fish, it is clear that in our case, liver was a better tissue than muscle to identify CTXs in fish for environmental assessment in the ecosystems. 

In the present work, the specimens which presented CTX-like toxicity were five moray eels corresponding to 3 species (*M. helena*, *M. augusti* and *G. unicolor*), grey triggerfish (*B. capriscus*), barred hogfish (*B. scrofa*), comb grouper (*M. fusca*) and bermuda sea chub (*K. sectatrix*). Even though CTXs have already been detected in almost all these species from the Atlantic Ocean [45,58,59]. The present study is the first report of CTX-like toxicity for *K. sectatrix* (Fam. Kyphosidae) from Macaronesia. *Kyphosus* species have been associated to CTXs in the South Pacific and Indian Ocean [60,61]. They are omnivorous and play a very important role as macroalgae consumers in temperate reefs [62,63]. Seven muscle samples of *K. sectatrix* from the Selvagens Islands were analysed previously, two in Costa et al. [45] by LC-MS/MS and five by Costa et al. [59] using neuro-2a CBA and LC-MS/MS, and no CTX analogues or CTX-like toxicity were detected in this species. In addition, in the current paper, the CTX-like toxicity is analysed for the first time for *C. labrosus* (grey mullet) from the Desertas Islands, and the two individuals tested were not toxic. As the *Kyphosus* genus, the genera *Chelon*, *Bodianus*, *Mycteroperca*, *Sparisoma* and *Balistes* are inclusive among ciguateric species in the Pacific and Caribbean Sea [60]. In the present study, the *B. scrofa*, *B. capriscus* and *M. fusca* were toxic as in the previous studies [45,49,59] even though that the extraction protocols were different from the one used in the current study and included an additional SPE cleaning step. Nevertheless, in the present study, the liver of *S. cretense* from the Desertas Islands did not exhibit CTX-like toxicity. This was not in accordance with Costa et al. [59], where fourteen muscles of *S. cretense* from the Selvagens Islands were analysed using neuro-2a CBA, and the CTX-like toxicity levels ranged from 0.006 to 0.04 µg of CTX1B equiv·kg^−1^.

As previously mentioned, the most toxic extracts for liver and muscle were those of moray eels (Table 1). This is in accordance with the characteristics of moray eels, which are benthic and large carnivorous species. Considering that moray eels have a sedentary behaviour, presence of CTXs in moray eels is a good indicator of CP risk in the area. In the literature, moray eels are the classical species which accumulate high quantities of CTXs. In fact, moray eels have been related to numerous CP cases [20]. Neuro-2a quantifications of CTX-like toxicity in the liver of the moray eels of the present study were comparable to levels in the muscle of moray eels from the Canary Islands [58]. The CTX-like toxicity and weight of all fish in the current study did not show an evident pattern. Nonetheless, when the toxicity of only the specimens of G. unicolor was plotted against their weight in Appendix A, a positive linear correlation was obtained (R^2^ = 0.79). Exemplars of this genus from the Pacific Ocean has already shown a positive correlation between CTX-like toxicity and weight [54].

In this study, six fish specimens from the Desertas Islands showed CTX-like toxicity. This is relevant because the Desertas Islands are close to Madeira, approx. 26 km to the south-east. Most of the ciguateric fishes reported from the Madeira and Selvagens Archipelagos were found in the Selvagens Islands, which are 260 km south-east from Madeira [43]. However, in Diogène et al. [64], three fish out of forty-seven (*Dentex gibbosus* and two *Seriola dumerili*) from Madeira Archipelago showed CTX-like toxicity using neuro-2a CBA. Previously, Costa et al. [45] analysed nine fish muscle samples from Madeira Island and none of them were positive by LC-MS/MS. On the other hand, CTXs were found in five out of eleven fish samples from the Selvagens Archipelago.

The European Regulations ban placing products containing CTXs in the market [65], but presently, there are no official methods to detect CTXs or a prescribed legal limit of CTXs in tissues. However, there is a guideline for maximum levels of 0.01 µg of CTX1B equiv·kg^−1^ and 0.1 µg of C-CTX1 equiv·kg^−1^ in tissue by the FDA [41] and the maximum concentration of 0.01 ug equiv. CTX1B kg^−1^ of fish recommended by European Food Safety Authority [42] to cover all CTX-group toxins that could be present in fish. In the present study, the estimated quantifications of CTXs in the positive samples for CTX-like toxicity were equal or higher than the established level of 0.01 µg of CTX1B equiv·kg^−1^. 

The neuro-2a CBA provides composite toxicity expressed in CTX1B equiv·kg^−1^ and does not provide information related to the specific CTXs present. In this study, the presence of two CTX analogues (C-CTX1 (*m*/*z* 1141.6305 [C_62_H_92_O_19_+H]^+^), dihydro-CTX2 (*m*/*z* 1115.5611 [C_60_H_84_O_18_+Na]^+^)), and a putative new CTX analogue with *m*/*z* 1127.6023 [M+NH_4_]^+^ were detected in muscle and liver extracts in three moray eels (*G. unicolor*) (codes 25, 26 and 27) from the Selvagens Islands by LC-HRMS. These three CTX compounds and gambieric acid A were also confirmed in the liver of one *M. augusti* (code 35) from the Desertas Islands by LC-HRMS. C-CTX1 has been detected before in samples from the Atlantic Ocean by Costa et al. [45], Estévez et al. [49], Pérez Arellano et al. [33], and Sánchez-Henao et al. [58], and their empirical formula was described by Abraham et al. [17] and Soliño et al. [53]. CTX2 has been found several times in fish and in moray eels (*Lycodontis javanicus*, Muraenidae) from Tarawa in the Republic of Kiribati (central Pacific Ocean) [6,66], albeit it is the first time that the putative CTX analogue corresponding to dihydro-CTX2 (C_60_H_84_O_18_) has been detected. It is necessary to be aware that C-CTX1 has not been detected in microalgae, and it is suggested that C-CTX1 is a CTX analogue resulting from fish metabolism. On the other hand, CTX2 was detected for the first time by Yogi et al. [67] in the toxin profile of a *Gambierdiscus toxicus* collected at Rangiroa Atoll of French Polynesia. Although C-CTX1, CTX2 and CTX3 jointly with CTX1B share the same mechanism of action [57,68], the potency among C-CTX1 and the Pacific congeners is suggested to be different. C-CTX1 is described as 10-fold less toxic than CTX1B by mouse bioassay (MBA) [69].

Gambieric acid A was found for the first time from *G. toxicus* isolated in the Gambier Islands in the Pacific Ocean [70]. Like CTXs, gambieric acids bind to the VGSCs [71]. However, gambieric acid A has not been related with CP, since a dose of 1 mg·kg^−1^ was not toxic in mice via intraperitoneal injection. Another gambieric acid, gambieric acid D, was found in fish for the first time by our group, specifically in a shark involved in a fatal food poisoning in the Indian Ocean [72]. This is the second time a gambieric acid has been detected in fish.

A putative CTX analogue (C-CTX-1109) corresponding to *m*/*z* 1127.6023 [M+NH_4_]^+^ and *m*/*z* 1132.5577 [M+Na]^+^ was also detected in the four liver and the two muscle samples of the fish analysed by LC-HRMS. It presents a larger retention time than CTX1B on the C_18_ column and a similar isotopic pattern in comparison to CTX1B main signals *m*/*z* 1128.6102 [M+NH_4_]^+^ and 1133.5656 [M+Na]^+^ and an experimental M+1/M ratio corresponding to a CTX analogue. However, it was not possible to identify a CTX structural formula for this compound. These CTX analogues may be the result of fish metabolism.

In accordance with the literature [44,45,46,47,49], our data suggest that the Caribbean ciguatoxin-1 (C-CTX1) would be responsible for ciguatera in the Selvagens and Desertas Islands. We observed a similar CTX profile in comparison to previous studies. Otero et al. [44] described the presence of C-CTX1, CTX3C and CTX1B and a CTX-analogue *m*/*z* 1040.6 in both fishes (*S. dumerili* and *S. fasciata*) from the Selvagens Islands. Costa et al. (2018) [45] confirmed and quantified C-CTX1 in four different species from the Selvagens Islands (*E. marginatus*, *M. fusca*, *B. scrofa* and *B. carpiscus*). Estevez et al. [47] described the presence of C-CTX1, C-CTX-1157 and C-CTX-1127 in a *S. fasciata* from the Selvagens Islands. None of the previous studies found gambieric acid A and dihydro-CTX2 in the profile of fishes from the Selvagens Islands.

Table 2 summarizes information found in the literature on CTX-positive fish samples from the Madeira and Selvagens Archipelagos, jointly with the results of the current study. The first CP episodes in Portugal were related to *Seriola* spp. caught in the Selvagens Archipelago and CTX analogues were confirmed by UPLC-MS in two individuals, *S. dumerili* and *S. fasciata* from the Selvagens Islands [44]. It is important to note that results on CTX-like toxicity of Otero et al. [44] cannot be strictly compared with the results of the current study since the estimations in Otero et al. [44] were performed using cerebellar granule cells (CGN), which is another functional assay based on cell electrophysiology, and also another standard (CTX3C) was used (Table 2). Afterwards, Caillaud et al. [36] detected CTX-like toxicity in two individuals of *S. fasciata* from the Selvagens Islands (Table 2), one of which was later confirmed to contain C-CTX1 by Estévez et al. [47]. The neuroblastoma assay implemented by Caillaud et al. [36], was comparable to the assay of the current study. CTX-like toxicity quantifications of *Seriola* spp. from Caillaud et al. [36] were much higher, over 100-fold higher than the most toxic fish of the present study. It is important to note that toxin contents can vary depending on the individual, species and the geographical location, year and season of capture.

## 4. Materials and Methods

### 4.1. Reagents and Equipment

CTX1B (also known as P-CTX-1), 52-epi-54-deoxy CTX1B (known as P-CTX-2 and CTX2) and 54-deoxy CTX1B (also known as P-CTX-3 and CTX3) standard solutions were provided by Prf. Richard J. Lewis (The Queensland University, Brisbane, Australia). CTX1B standard was used for the CBA and the LC-HRMS analysis. 52-epi-54-deoxy CTX1B and 54-deoxy CTX1B standards were used only for LC-HRMS analysis. Neuroblastoma murine cells (Neuro-2a) were purchased from ATCC LGC standards (Manassas, VA, USA). Poly-L-lysine, foetal bovine serum (FBS), L-glutamine solution, ouabain (O), veratridine (V), phosphate buffered saline (PBS), penicillin, streptomycin, RPMI-1640 medium, sodium pyruvate, thiazolyl blue tetrazolium bromide (MTT) were purchased from Merck KGaA (Darmstadt, Germany). Dimethyl sulfoxide (DMSO) and absolute methanol were purchased from Honeywell (Badalona, Spain) and Chemlab (Zedelgem, Belgium), respectively. Ultrasonic cell disrupter (Watt ultrasonic processor VCX750, Sonics, Newtown, CT, USA) and the Syncore^®^ Polyvap evaporator were purchased from Izasa Scientific (Alcobendas, Spain) and Büchi Syncore (Flawil, Switzerland), respectively. Automated plate spectrophotometer was purchased from Synergy HT Biotek elisa reader, Agilent, (Santa Clara, CA, USA).

### 4.2. Fish Samples

Seventeen specimens of nine species of fish were collected between October 2013 and December 2014 from several locations in the coastal waters of the Madeira and Selvagens Archipelagos, specifically in the Desertas Islands and Selvagens Islands (Table 3). Fishes were obtained in collaboration with the Parque Natural, Direção Regional das Pescas and local fishermen. For each individual, the following data were recorded: total length, and total fresh weight when caught (Table 3), and GPS coordinates and depth of capture (Appendix A). After capture, individuals were frozen and shipped to IRTA’s facilities. 

### 4.3. Identification of Fish by COI Sequencing 

Species were identified morphologically. In addition, for selected samples where species identification was not confirmed, a genetic analysis was performed using DNA sequences obtained by PCR of the cytochrome oxidase subunit I (COI) gene. Primers from previously published work [52] were used for the amplification of a fragment of this gene. The purified product was sent for bi-directional sequencing using the same primers as those used in the amplification. A BLAST analysis was performed and those samples for which an identity score of less than 95% was obtained, a further phylogenetic analysis was performed by Maximum Likelihood using MEGA ver7 [74].

### 4.4. Extraction Protocols

Fishes were dissected and muscle and liver of seventeen specimens were obtained (10 ± 0.1 g), each tissue was weighted in 50 mL Falcon tubes. Fish tissues were kept at −20 °C until toxin extraction. In order to perform the extraction, the sample was heated at 70 °C for 10 min in a water bath. Then, an acetone homogenization (2 mL g^−1^ wet weight of tissue) was performed with an Ultra-Turrax blender (IKA, T25 Basic, Staufen, Germany) for 3 min. The homogenate was centrifuged at 3000× *g* for 15 min at room temperature, and the supernatant was recovered and passed through a 0.22 μm PTFE filter (Whatman, Sigma-Aldrich, Merck, Madrid, Spain). The acetone homogenization was repeated twice. The supernatants were pooled and evaporated in a Syncore^®^ Polyvap evaporator (Büchi R-200, Flawil, Switzerland) at 40 ± 5 °C. After these steps, two different extraction protocols were followed based in the most common extraction protocols used in the literature.
Protocol-A

After the evaporation of the acetonic phase of all fish muscle, the aqueous residues were further extracted according to protocol-A based on Lewis et al. (2003). The aqueous extract was dissolved in 5 mL methanol:water (9:1, *v*:*v*) and partitioned twice with 5 mL *n-*hexane. The hexane layers were removed, and the aqueous methanol phase was evaporated with the Syncore^®^ Polyvap evaporator at 40 ± 5 °C until dryness. After that, the dried extract was dissolved in 5 mL ethanol:water (1:3, *v*:*v*) and partitioned twice with 5 mL diethyl ether. Diethyl ether fractions were pooled, dried, and the resulting residue was suspended in 4 mL of methanol and kept at −20 °C until analysis with the neuro-2a CBA.
Protocol-B

An alternative extraction protocol was implemented in all fish muscle and liver samples following protocol-B based on Yogi et al. [6] and modified by Dr. Jean Turquet [72]. After drying the acetonic phase, the aqueous residue was adjusted with Milli-Q water to 4 mL and diethyl ether was added, mixed by vortex, to initiate a water:diethyl ether partition (1:4, *v*:*v*) and kept at room temperature for 24 h. Afterwards, the diethyl ether phase was recovered and dried under N_2_ gas (Turbovap, Zymark corp, Hopkinton, MA, USA) at 40 °C. The dried extract was dissolved in methanol: water (8:2, *v*:*v*) and the extract was partitioned three times with *n-*hexane (1:2, *v*:*v*). The hexane layers were discarded, and the methanolic phase was dried with the evaporator at 60 ± 5 °C. Finally, the resulting residues were re-dissolved in 4 mL of HPLC-grade methanol and preserved at −20 °C until analysis with the neuro-2a CBA and subsequent confirmatory analysis using LC-HRMS.

### 4.5. CTX-like Toxicity Evaluation by Neuro-2a CBA

The evaluation of CTX-like toxicity was performed using the neuro-2a CBA following the procedure described in Caillaud et al. [36]. Briefly, neuro-2a cells (cell line CCL-131, ATCC LGC standards, Manassas, VA, USA) were seeded in 96 well-plates at 40,000 cells well^−1^. After 24 h, ouabain and veratridine (O/V) were added to a final concentration at 140 µM and 14 µM respectively, and 10 μL of each sample (serial dilutions of extract or standard, previously evaporated and resuspended in 5% FBS RPMI medium) was added to each well in triplicate. Tested concentrations ranged between 0.2 to 12 pg·mL^−1^ for CTX1B, 25 to 200 mg equiv·mL^−1^ of muscle and 1.88 to 50 mg equiv. mL^−1^ of liver. In each day of experimentation, a calibration curve was obtained with CTX1B (standard) which was tested in parallel to the samples to calibrate the experiment. Controls consisted of neuro-2a cells exposed to 5% FBS and not to extracts or standard in both conditions (O/V− and O/V+). All conditions were tested in triplicate. After 24 h of incubation, the neuro-2a cell viability was measured following the MTT colorimetric assay [75]. CTX-like activity of the extracts was estimated according to the standard dose-response curve following Caillaud et al. [36] and was expressed as µg equiv. of CTX1B kg^−1^ of tissue. Dose response curves were obtained using SigmaPlot 14.0 (Systat Software, San Jose, CA, USA). The limit of quantification (LOQ) of the method was defined as the amount of CTX1B that causes 20% of inhibition of cell viability (IC_20_) in O/V+ and that can be expressed according to the maximum concentration of tested tissue that does not affect the viability of neuro-2a cells under O/V− conditions [76].

### 4.6. LC-HRMS Analysis

For four moray eels, the four most toxic liver extracts (eels with codes 25, 26, 27 and 35 from Table 1) and two fish muscle extracts (eels with codes 26 and 27 from Table 1) as determined by neuro-2a CBA, an analysis of CTXs was conducted with LC-HRMS. For the analysis of CTXs by LC-HRMS an Orbitrap-Exactive HCD (Thermo Fisher Scientific, Bremen, Germany) mass spectrometer was used equipped with a heated electrospray source (H-ESI II), a Surveyor MS Plus pump and an Accela Open AS auto-sampler kept isothermal at 15 °C (Thermo Fisher Scientific, San Jose, CA, USA).

The chromatographic separation was performed on a reversed-phase Hypersil Gold C_18_ (50 mm × 2.1 mm, 1.9 µm) (Thermo Fisher, Scientific, Bremen, Germany) at a flow rate of 250 µL min^−1^. Mobile phase A was water and B was acetonitrile/water (95:5), both containing 2 mM ammonium formate and 0.1% formic acid. The gradient elution program for the analysis was: 30% B 1 min, 30–40% B 2 min, 40–50% B 1 min, 50–90% B 5 min, 90% B 3 min and return to initial conditions to re-equilibrate (2 min 30% B) and maintain these initial conditions (30% B) for 11 min. A 20-µL injection volume was used. The total duration of the method was 25 min. The analyses were carried out in positive electrospray ionization (ESI +) mode as described in Diogène et al. [72]. The final parameters were a spray voltage of 4.0 kV, capillary temperature of 275 °C, heater temperature of 300 °C, sheath gas flow rate of 35 psi, and auxiliary gas flow rate of 10 (arbitrary units). A capillary voltage of 47.5 V, tube lens voltage of 186 V, and skimmer voltage of 18 V were used. Nitrogen was employed as sheath, auxiliary, and collision gas. The mass range was *m*/*z* 400–1500 in full scan acquisition mode. The resolution was 50,000 (*m*/*z* 200, FWHM) at a scan rate of 2 Hz, and the automatic gain control was set as “balanced” (1 × 10^6^) with a maximum injection time of 250 ms. The data was processed with Xcalibur 2.2 SP1 software (Thermo Fisher Scientific, Bremen, Germany).

The peaks were extracted from the chromatogram using the sum of exact mass of [M+H]^+^, [M+NH_4_]^+^ and [M+Na]^+^ diagnostic ions with the ±10 ppm mass accuracy extraction window. The results were expressed as the sum of the three signals to improve sensitivity and harmonize the results regardless of the different relative intensity obtained for the three ions for each CTX and for several matrices. In addition to retention time, HRMS and mass accuracy parameters for accurate mass measurements (AMM), in the present study, to be confident in the identification and the proposed elemental formula, the following restrictive criteria were applied: elements considered were restricted in accordance with CTXs molecular formula and adduct signals [C: 55 to 70, H: 64 to 110, O: 11 to 25, N: 0 to 1, and cations (Na): 0 to 1]; the isotopic pattern was matched to theoretical in silico approach and the charge, the ring double bond equivalents (RDBEs) and nitrogen rule were taken into account. Additionally, the mono-isotopic pattern (M+1 ion) of these signals was used to assist in the further confirmation of the toxin identity as a supplementary identification point. Therefore, in total four diagnostic signals were used for toxin identification. Mass accuracy criteria was ≤ 8.5 ppm. The relative ion intensities between the main signal ([M+H]^+^, [M+NH_4_]^+^ or [M+Na]^+^) and their M+1 ions were calculated and matched taking into account a tolerance of 30% according to the EU Commission SANCO/12571/2013 guidance document. The characteristic isotopic pattern, M+1/M ion ratio, is a robust criterion that for all described CTXs is a theoretical value of 0.6–0.7. The combination of high resolution, mass accuracy and restrictive criteria was crucial for the identification of both targeted and unknown compounds.

## 5. Conclusions

The selection of the extraction protocol is crucial to obtain and identify toxins from fish matrices. In the fish used in this study from the Madeira and Selvagens Archipelagos, the protocol-B described by Yogi et al. [67] seems to be more suitable for recovery of CTX-like compounds than the protocol-A based on Lewis et al. [77]. As observed in previous studies, the combination of the neuro-2a CBA as a screening assay for CTX-like toxicity followed by CTX profile determination with LC-HRMS can be a good method for CTX evaluation. The HRMS full scan methodology has proved to be imperative for the reliable detection of putative CTXs, which are not monitored by targeting strategies such as MS/MS. Thirteen out of seventeen fish specimens from the Madeira and Selvagens Archipelagos presented CTX-like toxicities higher than the guidance level of 0.01 µg of CTX1B equiv·kg^−1^. Although the liver is a tissue that is not often consumed, it is an interesting organ to conduct risk assessment studies and to evaluate the presence of CTX-like compounds in fish, since levels of CTXs are higher than in muscle, according to our study. The detection of CTX-compounds in the liver can contribute as an early warning strategy to describe better the transfer and biotransformation of CTXs in the food webs in a specific area. 

The present data will benefit CP risk assessment in the region since they are bringing data on the presence of CTXs in different species of fish. Further systematic studies on a larger number of fish elected according to consumer’s habits can significantly improve risk assessment of ciguatera in the region. Among the ciguateric fish identified by neuro-2a CBA, six specimens were from the Desertas Islands and seven were from the Selvagens Islands. The highest toxicity was found in the liver of a *M. augusti* specimen (fish 35) from the Desertas Islands. Three CTX analogues (C-CTX1, dihydro-CTX2 and a putative new CTX *m*/*z* 1127.6023) and gambieric acid A were detected by LC-HRMS in some of the analysed samples, for example in the liver of fish with code 35, *M. agusti* from the Desertas Islands. Therefore, the Desertas Islands constitute a new area where ciguateric fish could be harvested and associated with CP cases. 

Liquid chromatography coupled to high-resolution mass spectrometry is an emerging and very attractive approach that combines qualitative and quantitative analyses, minimising the matrix effect, and also reducing the inaccuracies (false positives and negatives). The combination of high resolution, mass accuracy and restrictive criteria was crucial for identifying both targeted and unknown CTX compounds. Further work should be performed to obtain the structural characterization of these putative CTX compounds.

## Figures and Tables

**Figure 1 marinedrugs-20-00236-f001:**
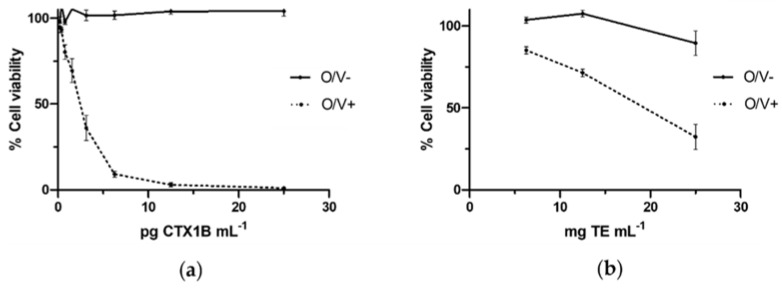
Dose-responses curves of neuro-2a cells to CTX1B or a liver fish extract, in the absence of ouabain and veratridine (O/V−) and the presence of ouabain and veratridine conditions (O/V+): (**a**) Dose-response curve of neuro-2a cells after the exposition to CTX1B standard; (**b**) Dose-response curve of neuro-2a cells after the exposure to liver fish extract (specimen with code 35). Data represent the mean and SD of three replicates on the same day of experimentation.

**Figure 2 marinedrugs-20-00236-f002:**
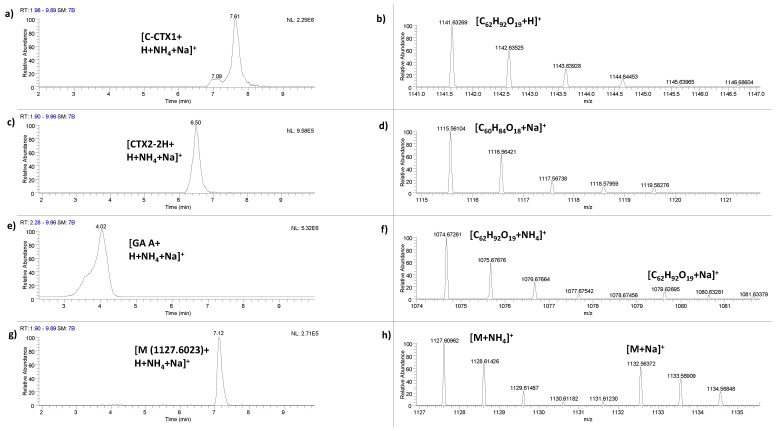
Evidence for the presence of ciguatoxins (CTXs): (**a**) Extracted ion chromatogram of C-CTX1 Σ([M+H]^+^ + [M+NH_4_]^+^ + [M+Na]^+^) in liver of fish 26 and (**b**) HRMS exact mass spectra of C-CTX1 [M+H]^+^ at *m*/*z* 1141.6305; (**c**) Extracted ion chromatogram of dihydro-CTX2 Σ([M+H]^+^ + [M+NH_4_]^+^ + [M+Na]^+^) in liver of fish with code 35 and (**d**) HRMS exact mass spectra of dihydro-CTX2 [M+Na]^+^ at *m*/*z* 1115.5611; (**e**) Extracted ion chromatogram of gambieric acid A Σ([M+H]^+^ + [M+NH_4_]^+^ + [M+Na]^+^) in liver of fish with code 35 and (**f**) HRMS exact mass spectra of gambieric acid A [M+NH_4_]^+^ at *m*/*z* 1074.6726 and [M+Na]^+^ at *m*/*z* 1079.6278; (**g**) Extracted ion chromatogram of putative analogue CTX (C-CTX-1109) Σ([M+H]^+^ + [M+NH_4_]^+^ + [M+Na]^+^) in liver of fish 25 and (**h**) HRMS exact mass spectra of a putative CTX analogue [M+NH_4_]^+^ at *m*/*z* 1127.6023 and [M+Na]^+^ at *m*/*z* 1132.5577.

**Table 1 marinedrugs-20-00236-t001:** CTX-like toxicity evaluation using the neuro-2a CBA of samples (muscles and livers) extracted with protocol B. Quantifications and LOQ are expressed as µg CTX1B equiv·kg^−1^ ± SD. Fish are in order of decreasing toxicity in liver.

			MUSCLE		LIVER	
Code	Species	Location	CTX-like Toxicity (µg CTX1B equiv·kg^−1^)	LOQ (µg CTX1B equiv·kg^−1^)	CTX-like Toxicity (µg CTX1B equiv·kg^−1^)	LOQ (µg CTX1B equiv·kg^−1^)
35 *	*M. augusti*	Desertas Islands	<LOQ	0.013	0.270 ± 0.121	0.044
27 *	*G. unicolor*	Selvagens Islands	0.034 ± 0.006	0.029	0.216 ± 0.051	0.011
25 *	*G. unicolor*	Selvagens Islands	<LOQ	0.005	0.212 ± 0.125	0.041
26 *	*G. unicolor*	Selvagens Islands	0.039 ± 0.001	0.002	0.187 ± 0.023	0.082
39	*M. helena*	Desertas Islands	0.083 ± 0.014	0.026	0.158 ± 0.022	0.044
34	*M. augusti*	Desertas Islands	0.065 ± 0.000	0.013	0.067 ± 0.022	0.044
14	*B. capriscus*	Selvagens Islands	<LOQ	0.010	0.060 ± 0.000	0.030
38	*M. helena*	Desertas Islands	<LOQ	0.013	0.050 ± 0.008	0.044
28	*G. unicolor*	Selvagens Islands	<LOQ	0.153	0.034 ± 0.000	0.004
6	*B. scrofa*	Selvagens Islands	<LOQ	0.016	0.015 ± 0.001	0.002
1	*M. fusca*	Desertas Islands	<LOQ	0.007	0.014 ± 0.000	0.008
13	*K. sectatrix*	Selvagens Islands	<LOQ	0.015	0.011 ± 0.008	0.003
37	*M. helena*	Desertas Islands	<LOQ	0.030	0.010 ± 0.002	0.004
2	*C. labrosus*	Desertas Islands	<LOQ	0.020	<LOQ	0.008
29	*C. labrosus*	Desertas Islands	-	-	<LOQ	0.047
31	*S. cretense*	Desertas Islands	-	-	<LOQ	0.178
36	*M. helena*	Desertas Islands	-	-	<LOQ	0.088

*: liver (25, 26, 27 and 35) and muscle (26, 27) of these individuals were also analysed by LC-HRMS.

**Table 2 marinedrugs-20-00236-t002:** Literature review including the present study on toxic fish from the Madeira and Selvagens Archipelagos including the Desertas (D) and Selvagens Islands (SI). n/a: weight not available; id: identification and confirmation without quantitative data.

Species	Weight of Specimen (kg)	Location (year)	Tissue	Toxicity According to CBA (µg equiv·kg^−1^)	Toxin Concentration by LC-MS Methods (µg equiv·kg^−1^)	Reference
*Balistes capriscus* (Grey triggerfish)	2.0	SI (2017)	Muscle		0.03 (C-CTX1)	[45]
	0.5–2.6	SI (2018)	Muscle		0.09 (C-CTX1)	[59]
	0.99	SI (2014)	Liver	0.060 (CTX1B)		This study
*Bodianus scrofa* (Barred hogfish)	1.6	SI (2017)	Muscle		0.11 (C-CTX1)	[45]
	0.8	SI (2017)	Muscle		0.06 (C-CTX1)	[45]
	n/a	SI (2018)	Muscle		id: C-CTX1	[49]
	1.1–3.0	SI (2018)	Muscle	0.04–0.75 (CTX1B)	0.08–0.48 (C-CTX1)	[59]
	2.4	SI (2014)	Liver	0.015 (CTX1B)		This study
*Diplodus cervinus* (Zebra seabream)	2.8	SI (2018)	Muscle	0.37 (CTX1B)	id: C-CTX analogue 1157 *m*/*z*	[59]
*Epinephelus marginatus* (Dusky grouper)	19.5	SI (2016)	Muscle		0.05 (C-CTX1)	[45]
*Gymnothorax unicolor* (Brown moray)	1.69	SI (2013)	Liver	0.212 (CTX1B)	id: CTX analogue 1127.6 *m*/*z*, C-CTX1 analogue 1141.6 *m*/*z*	This study


	1.21	SI (2013)	Muscle	0.039 (CTX1B)	id: CTX analogue 1127.6 *m*/*z*, dihydro-CTX2 analogue 1115.6 *m*/*z*	This study


			Liver	0.187 (CTX1B)	id: CTX analogue 1127.6 *m*/*z,* C-CTX1 analogue 1141.6 *m*/*z,* dihydro-CTX2 analogue 1115.6 *m*/*z*	This study



	1.3	SI (2013)	Muscle	0.034 (CTX1B)	id: CTX analogue 1127.6 *m*/*z*, dihydro-CTX2 analogue 1115.6 *m*/*z*	This study


			Liver	0.216 (CTX1B)	id: CTX analogue 1127.6 *m*/*z*, C-CTX1 analogue 1141.6 *m*/*z*	This study


	0.72	SI (2013)	Liver	0.034 (CTX1B)		This study
*Kyphosus sectatrix* (Bermuda sea chub)	3.4	SI (2014)	Liver	0.011 (CTX1B)		This study
*Mycteroperca fusca* (Island grouper)	4.6	SI (2016)	Muscle		0.25 (C-CTX1)	[45]
	2.1	DI (2014)	Liver	0.014 (CTX1B)		This study
*Muraena augusti* (Black moray)	1.7	DI (2013)	Muscle	0.065 (CTX1B)		This study
			Liver	0.067 (CTX1B)		This study
	1.3	DI (2013)	Liver	0.270 (CTX1B)	id:C-CTX1, dihydro-CTX2 analogue 1115.6 *m*/*z,* CTX analogue 1127.6 *m*/*z*	This study


*Muraena helena* (Moray eel)	1.68	DI (2013)	Liver	0.010 (CTX1B)		This study
	3.23	DI (2013)	Liver	0.050 (CTX1B)		This study
	1.19	DI (2013)	Muscle	0.083 (CTX1B)		This study
			Liver	0.158 (CTX1B)		This study
*Pagrus pagrus* (Red porgy)	4.0	SI (2016)	Muscle		0.76 (C-CTX1)	[46]
*Seriola dumerili* (Greater amberjack)	70	SI (2009)	Muscle (tail)	37.3 (CTX3C)	53.76 (CTX3C)	[44]
					id: CTX3C, CTX analogue 1040.6 *m*/*z*, CTX analogue 1141.6 *m*/*z*	

			Muscle (head)	40.6 (CTX3C)	54.35 (CTX3C)	[44]
					id: CTX3C, CTX analogue 1040.6 *m*/*z*, CTX analogue 1141.6 *m*/*z*	

			Muscle (ventral)	5.1 (CTX3C)	33.29 (CTX3C)	[44]
					id: CTX1B, CTX3C, CTX analogue 1040.6 *m*/*z*, CTX analogue 1141.6 *m*/*z*	

			Muscle (mid)	41.7 (CTX3C)	53.37 (CTX3C)	[44]
					id: CTX1B, CTX3C, CTX analogue 1040.6 *m*/*z*, CTX analogue 1141.6 *m*/*z*	


			Liver	37.7 (CTX3C)	48.60 (CTX3C)	[44]
					id: CTX1B, CTX3C, CTX analogue 1040.6 *m*/*z*, CTX analogue 1141.6 *m*/*z*	
*Seriola fasciata* (Lesser amberjack)	30	SI (2009)	Muscle (tail)	40.6 (CTX3C)	35.2 (CTX3C)id: CTX1B, CTX3C, CTX analogue 1040.6 *m*/*z*, CTX analogue 1141.6 *m*/*z*	[44]



	63	SI (2012)	Muscle	6.23 (CTX1B)		[36]
	37	SI (2012)	Muscle	4.59 (CTX1B)		[36]
	37	SI (2008)	Muscle	1.4 (C-CTX1)	0.84 (C-CTX1)	[47]
*Seriola rivoliana* (Longfin yellowtail)	2.5–12.3	SI (2018)	Muscle	0.10 (CTX1B)		[59]
	n/a	SI (2008)	Muscle	0.17 (C-CTX1)	id: C-CTX1	[73]
*Serrianus atriacuda* (Blacktail comber)	0.2–0.3	SI (2018)	Muscle	0.006–0.02 (CTX1B)		[59]
*Sparisoma cretense* (Parrotfish)	0.4–0.8	SI (2018)	Muscle	0.006–0.04 (CTX1B)		[59]
*Sphyraena viridensis* (Yellowmouth barracuda)	2.2–5.9	SI (2018)	Muscle	0.22 (CTX1B)	0.14 (C-CTX1)	[59]

**Table 3 marinedrugs-20-00236-t003:** List of individuals captured between October 2013 and December 2014 in the coastal waters of the Madeira and Selvagens Archipelagos. Species were determined by morphological identification and molecular genetics. FB: feeding behaviour (C: carnivorous, O: omnivorous); G: gender (F: female, M: male); no det: undetermined.

Code	Common Name	Species	Location	Date of Capture	Weight (kg)	Length (mm)	FB	G
1	Comb grouper	*M. fusca*	Desertas Islands	25 September 2014	2.106	505	C	F
2	Grey mullet	*C. labrosus*	Desertas Islands	25 September 2014	1.972	572	O	M
6	Barred hogfish	*B. scrofa*	Selvagens Grande	24 May 2014	2.432	500	C	M
13	Bermuda sea chub	*K. sectatrix*	Selvagens Grande	25 May 2014	3.431	570	O	F
14	Grey triggerfish	*B.capriscus*	Selvagens Grande	25 May 2014	0.997	388	C	F
25	Brown moray	*G. unicolor*	Selvagens Grande	31 December 2013	1.688	894	C	no det
26	Brown moray	*G. unicolor*	Selvagens Grande	31 December 2013	1.210	838	C	no det
27	Brown moray	*G. unicolor*	Selvagens Grande	31 December 2013	1.305	812	C	no det
28	Brown moray	*G. unicolor*	Selvagens Grande	31 December 2013	0.721	717	C	no det
29	Grey mullet	*C. labrosus*	Desertas Islands	27 December 2013	1.896	545	O	M
31	Parrotfish	*S. cretense*	Desertas Islands	27 December 2013	0.412	307	O	F
34	Black moray	*M. augusti*	Desertas Grande	8 November 2013	1.669	917	C	M
35	Black moray	*M. augusti*	Desertas Grande	8 November 2013	1.397	838	C	M
36	Moray eel	*M. helena*	Desertas Grande	8 November 2013	1.339	893	C	no det
37	Moray eel	*M. helena*	Desertas Grande	8 November 2013	1.680	931	C	F
38	Moray eel	*M. helena*	Desertas Grande	8 November 2013	3.234	1070	C	M
39	Moray eel	*M. helena*	Desertas Grande	8 November 2013	1.193	856	C	M

## Data Availability

All data of the study are in the manuscript.

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
