# Peer review of "Identification of New CTX Analogues in Fish from the Madeira and Selvagens Archipelagos by Neuro-2a CBA and LC-HRMS"

_marinedrugs, 2022, doi:10.3390/md20040236_

Round 1

Reviewer 1 Report

Review of “Identification of new CTX analogues in fish from the Madeira and Selvagens Archipelagos by Neuro-2a CBA and LC-HRMS” by Àngels Tudó et al.

The article is well-written and makes an important contribution to the evolving understanding of the ciguatoxins (CTXs) present in fish in different regions of the world. The authors do an excellent job of reviewing the history of the toxins identified in fishes found in the eastern Atlantic and identified a previously unknown C-CTX structural variant. The authors also present a modified extraction protocol that improved the extraction efficiency of CTXs from fish tissues with no extra effort which other laboratories will likely adopt. Further, the results will be of general interest to both Harmful Algal Bloom scientists and to managers and public health officials responsible for protecting the public from ciguatoxin exposure. The article is recommended for publication after very minor revision.

Line

63 suggest “have increased” instead of have been increased

175 G. unicolor and M. augusti should be italicized

196 suggest deleting somehow in the phrase “…somehow measures the “degree  of unsaturation” of the molecule”.

237 suggest “The observed signals, however, do not agree…”

255 Gymnothorax and Muraena should be italicized

258 G. unicolor and M. augusti should be italicized

274 suggest “prior to the hexane cleaning.”

277-279 Combine with the previous paragraph.

278 the meaning of the phrase “to select the tissue to reveal” is ambiguous.  Perhaps “…indicate that it is important to survey different tissues (particularly liver tissues) to establish whether or not a given fish contains measurable CTXs concentrations.”?

280 The phrase “For the fish tested below” is ambiguous. Specific exactly which fish are being referenced.

302  M. helena, M. augusti and G. unicolor itallics (seems like upload program may have stripped italics from all the species names.

303, 306, 309, 313-316, 319, 321, 333, 341, 342, 358, 360, 365, 375, 394, 396, 397, 402, 404, 409, 413, 578, 581 italicize species names

313-314 the no toxicity statement is pretty weak given the sample size studied.

329 Neuro-2A quantifications of…

357 CTX or C-CTX analogue

The sections 281-286 and 296-300 overlap to some degree and could be combined or arranged.

390-391 In accordance with the literature [44-47, 49], we confirmed the Caribbean ciguatoxin-1 (C-CTX1) to be responsible for ciguatera from the Selvagens and Desertas Islands. Though probably true, this statement is not fully addressed using the MS data generated in this study. Realize the next suggestion is challenging given the lack of standards and differing sensitivity issues, but Is there a way that MS results for the various identified congeners could be converted to approximate CTX1B equivalents thereby showing the dominance of C-CTX1? If this could be accomplished it would be useful to show how much of the N2a activity could be accounted for?  Alternatively, the statement could be bolstered by including a small table showing the relative toxicity of the various congeners which documents the higher relative toxicity of C-CTX-1. 

Table 2 column heading why CBA instead or Neuro-2a or Neuro-2a CBA. Also, the concentration entry across from (Barred hogfish) is out of place.  0.48 C-CTX1 (id) instead of C-CTX1 (id) 0.48

Figure 1.  The phrase "..after the exposition of liver fish extract.." is not clear. After exposure to fish liver extracts?

423 Suggest adding the following stating after “…(The Queensland University, Australia). Note, these toxins, as well as other well characterized CTX compounds derived from Pacific fish or Gambierdiscus polynesiensis cultures share a common backbone ring structure.  Unlike these Pacific derived toxins, which a not denoted using a prefix, the CTXs with different structures isolated from the Caribbean (Atlantic) or Indian Ocean fishes are distinguished by adding the prefix C- or I-, respectively.”  This often taken as given knowledge, but would help readers not familiar with how the nomenclature of the toxins has evolved over the past couple of decades and who might not appreciate the difference between CTX-1 and C-CTX-1.

531  1,500   50,000 versus 1500 and 50 000?

Author Response

Line 63 suggest “have increased” instead of have been increased. We changed it. (line: 63)

175 G. unicolor and M. augusti should be italicized. We changed it and we have checked all italics.

196 suggest deleting somehow in the phrase “…somehow measures the “degree  of unsaturation” of the molecule”.

We have changed it. Now is: Additionally, the RDBE term measures the degree of unsaturation of the molecule (line:196).

237 suggest “The observed signals, however, do not agree…” Done (line 237)

255 Gymnothorax and Muraena should be italicized we checked all italics.

258 G. unicolor and M. augusti should be italicized. we checked all italics.

274 suggest “prior to the hexane cleaning.” We changed it (lines: 273-274)

277-279 Combine with the previous paragraph. We combined these paragraphs.

278 the meaning of the phrase “to select the tissue to reveal” is ambiguous.  Perhaps “…indicate that it is important to survey different tissues (particularly liver tissues) to establish whether or not a given fish contains measurable CTXs concentrations.”? Yes, it is true. We changed the phrase and also we arranged as it has been suggested below. (line: 279)

280 The phrase “For the fish tested below” is ambiguous. Specific exactly which fish are being referenced. Yes, it is true. We have referenced fish using the fish code. (line:276)

302  M. helena, M. augusti and G. unicolor itallics (seems like upload program may have stripped italics from all the species names. we have checked all italics.

303, 306, 309, 313-316, 319, 321, 333, 341, 342, 358, 360, 365, 375, 394, 396, 397, 402, 404, 409, 413, 578, 581 italicize species names

313-314 the no toxicity statement is pretty weak given the sample size studied. Yes, it is true. We propose slight modification of the sentence (lines:305-307).

329 Neuro-2A quantifications of… We changed it. (line 322)

357 CTX or C-CTX analogue We changed it.

The sections 281-286 and 296-300 overlap to some degree and could be combined or arranged. As we mentioned, we combined them. Lines: 282-293

390-391 In accordance with the literature [44-47, 49], we confirmed the Caribbean ciguatoxin-1 (C-CTX1) to be responsible for ciguatera from the Selvagens and Desertas Islands. Though probably true, this statement is not fully addressed using the MS data generated in this study. We understand that is true and we have changed the lines: 383-384.

Realize the next suggestion is challenging given the lack of standards and differing sensitivity issues, but Is there a way that MS results for the various identified congeners could be converted to approximate CTX1B equivalents thereby showing the dominance of C-CTX1? If this could be accomplished it would be useful to show how much of the N2a activity could be accounted for?  Alternatively, the statement could be bolstered by including a small table showing the relative toxicity of the various congeners which documents the higher relative toxicity of C-CTX-1. We don’t have information of the toxicity of other CTX analogues, just it is available for C-CTX1.  

Table 2 column heading why CBA instead or Neuro-2a or Neuro-2a CBA. Also, the concentration entry across from (Barred hogfish) is out of place.  0.48 C-CTX1 (id) instead of C-CTX1 (id) 0.48. There are other types of CBA that are not neuro-2a. We fixed the columns. 

Figure 1.  The phrase "..after the exposition of liver fish extract.." is not clear. After exposure to fish liver extracts? We rephrased it (line 152).

423 Suggest adding the following stating after “…(The Queensland University, Australia). Note, these toxins, as well as other well characterized CTX compounds derived from Pacific fish or Gambierdiscus polynesiensis cultures share a common backbone ring structure.  Unlike these Pacific derived toxins, which a not denoted using a prefix, the CTXs with different structures isolated from the Caribbean (Atlantic) or Indian Ocean fishes are distinguished by adding the prefix C- or I-, respectively.”  This often taken as given knowledge, but would help readers not familiar with how the nomenclature of the toxins has evolved over the past couple of decades and who might not appreciate the difference between CTX-1 and C-CTX-1. Regarding this suggestion, to include here additional information on structures of CTXs, we understand that this information is not critical in this section regarding Materials and Methods.

531  1,500   50,000 versus 1500 and 50 000? Done

Reviewer 2 Report

First of all, authors performed a very nice work! It is very important to clarify the ciguatoxin profile of ciguatera poisoning of northern Atlantic areas. The investigation of ciguatera poisoning of this area has been started lately compared to Pacific, Caribbean and Indian areas.

Authors introduced a combination of Neuro-2a assays and LC-HRMS analysis to screen the ciguatera fish or eels and to clarify the contents of ciguatoxins, respectively. This methodology worked very well to reveal the toxicity of the samples and find a new contents, which were dhydro-CTX2 and gambieric acid A.

The impact of this manuscript is big for researcher of ciguatoxins. This manuscript should be accepted. However, there are some issues listed below to be corrected before acceptance.

1, In identification of dihydro-CTX2 and gambieric acid A, the comparison of retention times between sample solution and authentic standard solution. I can understand that compound in the sample solution showed supporting data in HRMS analyses, but the possibility of structural isomer cannot be excluded.

2, Authors should check the manuscript for the punctuation, i.e. scientific name should be in italic, n for n-hexane should be in italic, etc.

3, In line 480, I could not understand the meaning of “43”.

4, In line 539, I could not understand the meaning of “ans”.

Author Response

1, In identification of dihydro-CTX2 and gambieric acid A, the comparison of retention times between sample solution and authentic standard solution. I can understand that compound in the sample solution showed supporting data in HRMS analyses, but the possibility of structural isomer cannot be excluded.

We agree with the reviewer, we have added “or structural isomer” in the identification of dihydro-CTX2 and gambieric acid A (lines: 215-222).

2, Authors should check the manuscript for the punctuation, i.e. scientific name should be in italic, n for n-hexane should be in italic, etc. we checked it.

3, In line 480, I could not understand the meaning of “43”. It is true, we changed it. It was a reference that was not well edited  (line 473)

4, In line 539, I could not understand the meaning of “ans”. It is true. we changed it (line: 533).

Round 2

Reviewer 2 Report

Authors are advised to change the expression "or structural isomer" into another words, for example, "unknown" or "UK-". The expression "or structural  isomer" does not mean that the authors identified the specific compound. The compounds name should not be used for the unknown compounds. Authors should then describe their spectral data and the structural characteristics of the compounds in the main text.

Author Response

Authors are advised to change the expression "or structural isomer" into another words, for example, "unknown" or "UK-". The expression "or structural  isomer" does not mean that the authors identified the specific compound. The compounds name should not be used for the unknown compounds. Authors should then describe their spectral data and the structural characteristics of the compounds in the main text.

At the suggestion of reviewer 2 in the first review, we have added “or structural isomer” in the identification of dihydro-CTX2 and gambieric acid A. They are not "unknown" or "UK-". As described in the manuscript, there are sufficient HRMS evidences (exact mass, isotopic pattern, RDBE, elements in use, nitrogen rule) to identify the signals as gambieric acid and dihydro-CTX2. But, as reviewer 2 indicate, we do not have the standard to confirm. Therefore it could also be a structural isomer, with the same formula molecular and isotopic profile. In particular, in the case of dihydro-CTX2, we cannot specify more because with the data obtained the position of the double bond cannot be specified.

So we would think it is better to include this term responding to the first review.